

# The development and test research of multi-channel Synchronous transient electromagnetic receiver

Fanqiang Lin [1,2], Xuben Wang [2,3], Kecheng Chen [1], Depan Hu [4], Song Gao [1], Xue Zou [1], and Cai Zeng [1]

[1]College of Information Science and Technology, Chengdu University of Technology, Chengdu 610059,China
[2]College of Geophysics,Chengdu University of Technology, Chengdu 610059,China
[3]Key Lab of Geo-Detection and Information Techniques of Ministry of Education, Chengdu 610059,China
[4]Geoenvironment Monitoring Institute in Chengdu, Chengdu 610071,China
**Correspondence:** Fanqiang Lin(linfq@cdut.edu.cn)

**Abstract.** As a result of the drastic reduction of shallow mineral resources, the exploitable potential and reserve of proven mines are insufficient, and the mineral resources in deep ground need to be more refinedly explored.There are some disadvantages of the existing instruments, such as few channels and slow sampling rate, etc. Therefore, multi-parameter transient electromagnetic instrument with synchronous receiving has been developed and tested. The instrument is composed of two controllers, embedded controller and programmable logic controller, which can provide diversified information combination. Under the grounding electrode source emission mode, the real-time synchronous transient electromagnetic acquisition system of six channels is achieved with 128k sampling rate . The data of the six channels are recorded in the full time range in the time domain. At the same time, experiments were carried out in laboratory, open areas and actual mine. Through data analysis, the measured data curves of the mining area are highly consistent with the existing geochemical exploration curves and geological profile.

## 1 Introduction

Transient electromagnetic method belongs to the active field source method of time domain electromagnetic method. The principle is that the conductive geological body existing underground produces induced eddy currents under the action of alternating electromagnetic fields, and the eddy current further generates secondary magnetic fields. In transient electromagnetic method, periodic bipolar waves are used as launch signals, and data acquisition is completed during the bipolar wave intermittence,thus effectively avoiding the coupling problem in the frequency domain electromagnetic method. The instrument collects and studies the decay characteristics of the intensity of each component, distribution, space and time of the secondary fields to deduce the underground anomalies. The transient electromagnetic signals contain very rich frequency components, which can easily penetrate low-resistivity geologic bodies, and therefore the actual exploration depth can be effectively improved.

Conventional transient electromagnetic (TEM) uses mostly magnetic source or electrode source mode. In the magnetic source mode TEM volume effect is relatively small,and it has high detection accuracy, but the effective detection depth is usually less than 500 meters, and the response to high resistivity geological bodies is not sensitive. However, LOTEM can increase the exploration depth up to 10 km. However, due to the large distance between the LOTEM transmission and reception,



the accuracy of the collected data is relatively low, as well as a poor signal to noise ratio(SNR) and increased possibility of interference on the way of long-distance transmission(Brunke et al.,2017). As a result, domestic scholars such as Xue Guoqiang proposed short offset transient electromagnetic (SOTEM) device method(Xue et al.,2013,Chen et al.,2016,Chen et al.,2017), the reduction of transceiver distance can both greatly enhance the exploration depth, but also effectively improve the

5 SNR. In recent years, it has a very broad application prospect in such fields as engineering environment, geological disaster investigation, underground gob area, groundwater, petroleum, coal and other non-metallic mineral exploration and advanced geological prediction.

## 2 Literature review

In 2010, internationally renowned geophysicist Zhdanov proposed the following directions for future electromagnetic explo-

10 ration instruments and methods in the 75th anniversary of "Geophysics": Multi-component emission, Multi-channel reception and Pseudo-seismic data collection (Qi et al.,2015,Ayuso et al.,2016). As the demand for resources and energy increases due to economic development, the depth is increased to the second depth space (500-2000m) on the basis of the existing exploration depth (500m depth range). The fine exploration within this range will be a long-term and heavy task to study new theories and methods and to develop equipment with higher adaptability to make up the lack of traditional exploration methods and

15 instruments.

China's overall theoretical level and engineering practice in the field of transient electromagnetic remain at the international level, but compared with foreign instrument, the domestic instruments still have some disadvantages in the technical indicators, which will be constraining China's original innovation and quantitative calculation of the bottleneck(Zhong et al.,2016,Zhang et al.,2017). With the development of electrical prospecting theory, instrument's design and development, many kinds of in-

20 strument products have been made. The main development directions are automation, intelligence, refinement and lightness. As a result, various institutions of higher learning and research institutes in China have introduced various forms of transient electromagnetic prospecting instruments in their decades of development. By comparing most of the transient electromagnetic surveys at home and abroad in recent 20 years, the main problems of the domestic transient electromagnetic instrument lie in that: some technical indicators are lagging behind; the function is insufficient; the consistency is low and so on. It can

be seen that the development of new acquisition devices, the transformation of new launch modes, acquisition methods, the development of new instruments and evaluation methods of electromagnetic methods, etc. are particularly important(Li et al.,2012). The core of this paper is to develop a receiver system for synchronous acquisition of multiple electromagnetic signals in transient electromagnetic prospecting to achieve multi-parameter and multi-channel synchronous reception. High-speed programmable logic devices are used to achieve high-level synchronization between channels. Transmitting current waveform

acquisition and multi-channel reception can be synchronized by using high-precision GPS timing unit, and programmable high-precision counter is adopted to further improve the synchronization accuracy.



## 3    Multi-channel receiver hardware and software design

### 3.1    Receiver framework

Receiving system consists of four parts, namely: FPGA unit, ARM unit, analog acquisition unit and power supply circuit.
System block diagram shown in Figure 1. Analog board channel numbers are 1/2, 3/4, 5/6 channels respectively, and each

5    analog board have two channels. The board contains the signal conditioning circuit, amplification and acquisition of all channels
which are completely independent, and multi-channel real-time synchronized acquisition is achieved.

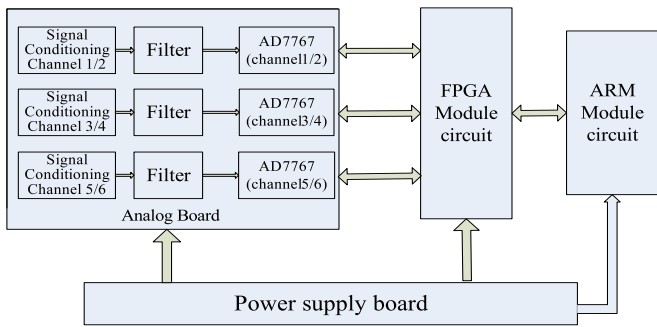

**Figure 1.** System block diagram of hardware

### 3.2    Design of analog circuit board

Analog board mainly includes signal conditioning (including: the first stage instrumentation amplifier, the second stage pro-

10    grammable amplifier, low-pass filter), protection circuit (positive and negative polarity of each input stage clamp diode), the
natural potential compensation circuit(Liu et al.,2016), single-end to differential circuit and analog to digital converter.

The first stage amplifier uses a dedicated high-precision, low-noise INA114, and using the AD5272 high-resolution digital
potentiometer to design the precision gain op-amp, to achieve high-precision amplification and range adjustment.The two
circuits are shown in Figure2 and Figure3.

It utilizes the PGA103 program-controlled amplifier, and a general IO port to complete the three level amplification ratio
adjustment.




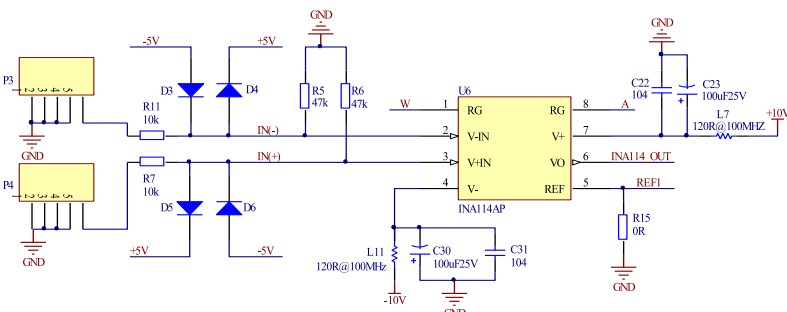

**Figure 2.** INA114 instrument amplifier circuit

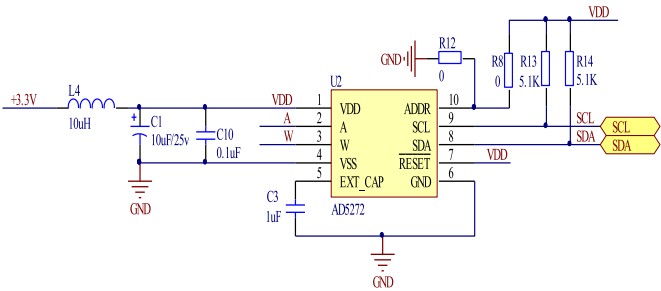

**Figure 3.** AD5272 Digital potentiometer

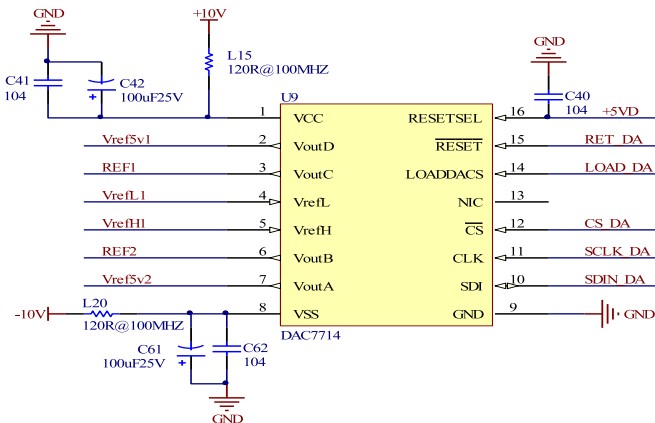

**Figure 4.** DAC7714 compensation and Vref circuit

Natural Potential Compensation uses the DAC7714, a 12-bit digital-to-analog converter with four output channels, which can provide natural potential compensation and full-scale voltage of both channels.The DAC7714 circuits is shown in Figure4.

Single-end to differential device is based on the needs of analog-digital converter settings, while the instrument uses the ADA4941. The system utilizes 24-bit AD7767, a 24-bit high resolution$\sum -\Delta ADC$ with over-sampling on each channel,to



reduce noise from the front end and the need for front-end anti-alias filter, and daisy chain technology to realize the multi-chip cascade connection for an efficient parallel synchronous acquisition (Liu et al.,2017).The two circuits are shown in Figure5 and Figure6.

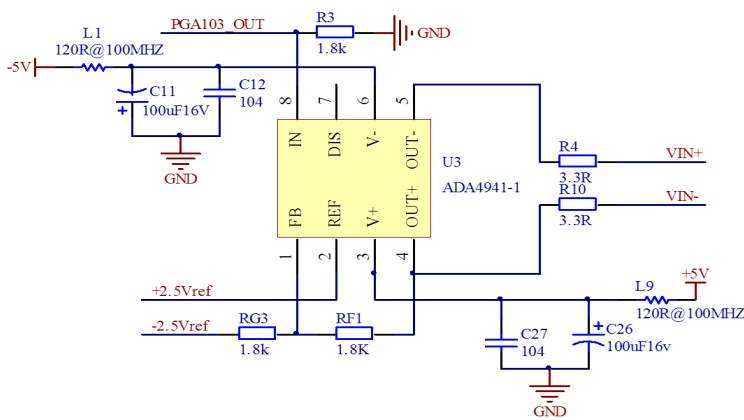

**Figure 5.** ADA4941 single to differential convertor circuit

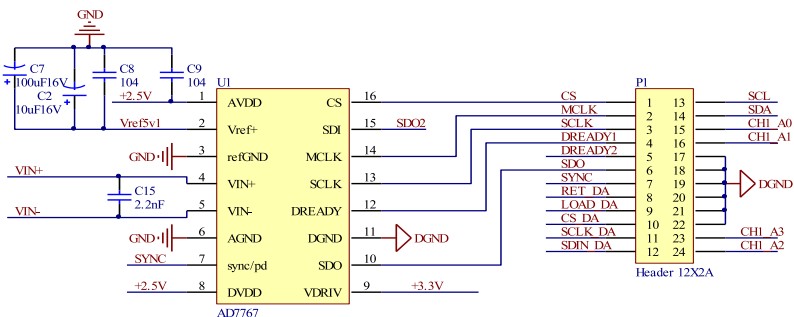

**Figure 6.** ADC7767 analog to digital convertor circuit

By adopting modular design, two AD7767 chips are cascaded, FPGA controller is used to control three analog circuit boards separately, and the overall data package and storage in real-time is achieved.

## 3.3 Design of digital logic controller board

Multi-channel synchronous receiver uses embedded controller and programmable logic controller (FPGA) as dual-controller

10 (Oballe-Peinado et al.2017) , which is flexible and widely used. FPGA is the core logic control part of the collection system, and



the synchronization between the various functional units can be easily achieved by its parallel processing function, by which a number of logic modules are driven by the system clock.This instrument uses the XC6SLX9 chip as the core acquisition controller,that belongs to the Spartan-6 series of well-known FPGA chip maker Xilinx.FPGA unit is mainly composed of SD card unit circuit, SRAM buffer circuit and the SPI interface which is used to communicate with STM32 controller, the

5  synchronous acquisition and control interface of six channel signals, filter frequency output interface, and GPS's second-pulse signal interface, and power interface.

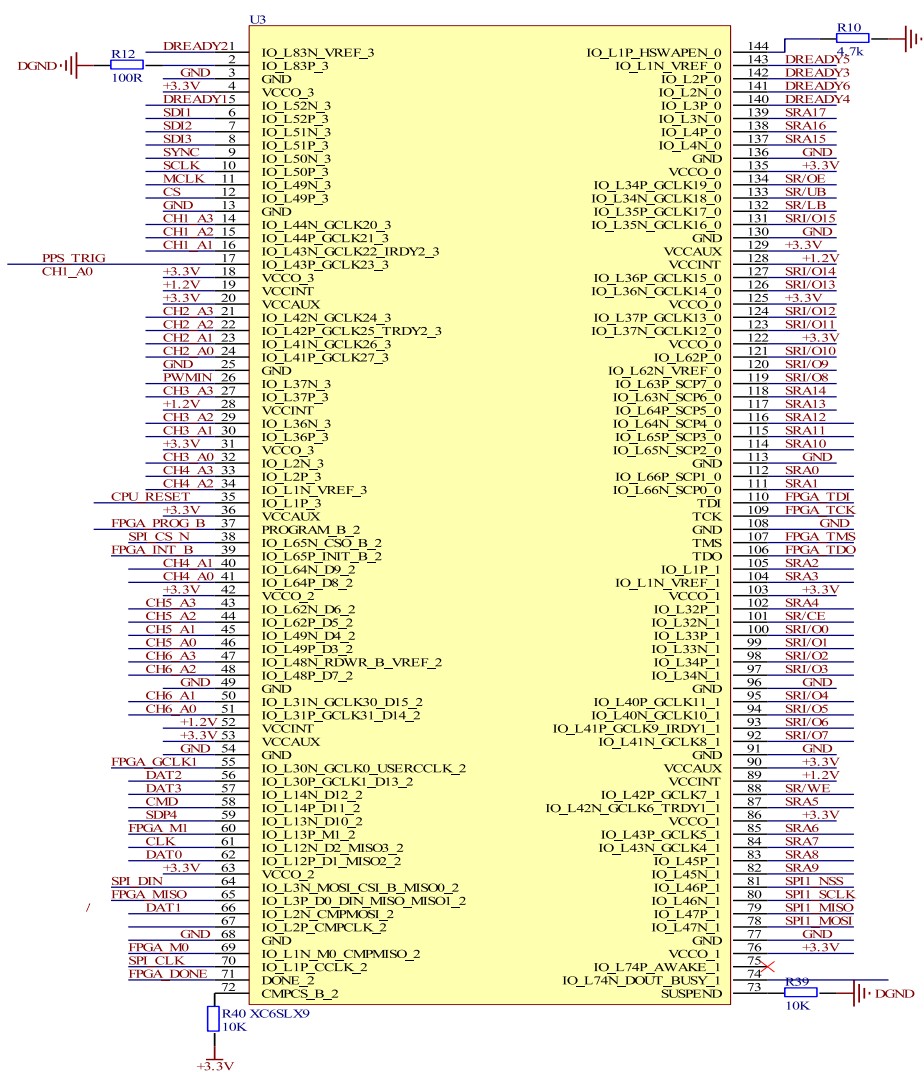

**Figure 7.** XC6SLX9 FPGA circuit diagram





The embedded control unit adopts the STM32F4 series of ST Microelectronics which is popular in industry. The embedded control part has completed the functions of keyboard and FPGA communication control interface, GPS location information collection and liquid crystal display. The embedded unit communicates with the FPGA controller by SPI interface. It also collects the GPS time information, such as latitude and longitude, etc. which is outputted by the GPS module through the serial

5   port, and simultaneously displayed on LCD. A 6*8 metrix keyboard is designed to set some parameters of the system. When the GPS satellite signal is locked, the receiving system can collect the data according to the preset instruction by start key. Both selecting the channel for collection and setting the channel on or off can be finished with the LCD and keyboard control in actual work.And general purpose IO port is used to control DAC7714 chip to achieve natural potential compensation and fine gain tuning.The logic core diagram is shown in figure7,and the STM32F407 diagram is shown in figure8.

**Figure 8.** STM32F407 circuit diagram



## 3.4   Design of high-precision linear power circuit

After comparing the multi-output power supply circuit board designed by the switching power supply module, a Low Drop Out linear power chip with high precision and high ripple rejection is selected(Joo et al.,2017,Duong et al.,2017). On the power supply board, three independent connecting plugs are used to supply power to each analog board(Ren et al.,2015). Here, the high-reliability TPS7A series of voltage regulators designed by Taxi Instrument which features wide input voltage range, low noise, and high supply ripple rejection is selected. The power board has 4 different voltage output: + 10V, -10V, + 5V, -5V for preamplifiers and filters. Another separate + 5V power supply for digital potentiometers on analog board and DAC7714, etc., and a 3.3 V Power provides digital logic power for Analog-to-Digital converter AD7767(Yun et al.,2017).The high precision and high power supply rejection rate diagram is shown in figure9.

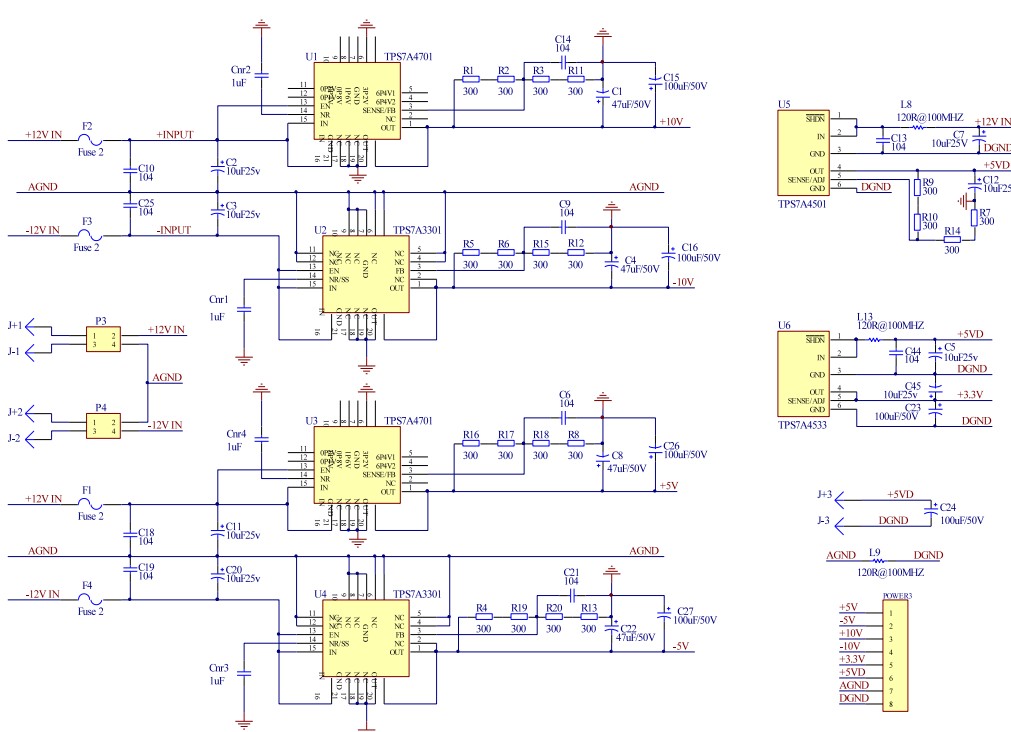

**Figure 9.** Linear power circuit diagram

## 3.5   Software design

The software mainly includes two parts: 1)FPGA acquisition control board program, which involves the use of 4-bit data transfer on the SD card storage, real-time communication with STM32, and data processing in data buffer area ,etc. 2)and





STM32 control board program, which includes the gain adjustment of the first stage preamplifier, the ratio setting of the second stage of programmable amplifier, and natural potential compensation control(Khomutov et al.,2017).

FPGA mainly controls the front-end analog-digital converter, data storage and interactions with STM32. The FPGA chip first loads the stored program file after power-on, reads the flash through the SPI interface, and then waits for the operation control
commands sent by the STM32, such as system initialization, parameter setting and sensor calibration(Wang et al.,2017). Time information and the channel numbers are stored in a package through FPGA program. Those information is stored in the first 8 bytes of each sector in SD card. The second pulse signal(PPS) provided by GPS timing unit updates the internal base time of FPGA in every two seconds. It can reduce the time cumulative error.

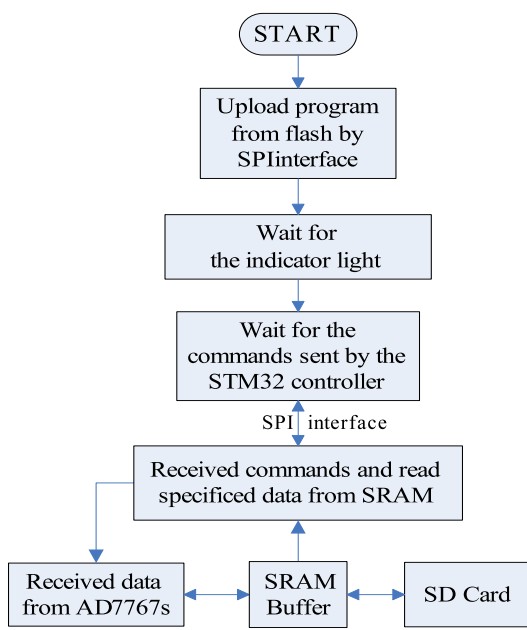

**Figure 10.** Flowchart of system program

Embedded controller is responsible for coordinating the operation of the entire system, such as the system gain adjustment of each channel, GPS time information reception and status monitoring, real-time display of acquired data,and interface communicating with FPGA controller(Zhang et al.,2017).

## 4  Performance test

Performance test of the receiver system mainly includes DC testing, AC testing and field testing(Ziolkowski et al.,2010,Zhou et
al.,2015).In the DC stability test, we first test the magnification range of each channel and the magnification consistency. Taking



**Table 1.** Magnification test of each channel

| Channel No. | Magnification mean |
|---|---|
| Channel 1 | 94.42 |
| Channel 2 | 94.46 |
| Channel 3 | 94.39 |
| Channel 4 | 94.68 |
| Channel 5 | 94.32 |
| Channel 6 | 94.59 |

the first channel as an example, as shown in Figure 11 below, the input signal is a blue curve with a range of ± 50mV. The red curve is the magnification curve of the first channel when each input signal is calculated. Under the same test conditions, the six channels are connected in parallel,and the same input signals are received by six channels at the same time.The average magnification times of each channel are shown in Table 1.

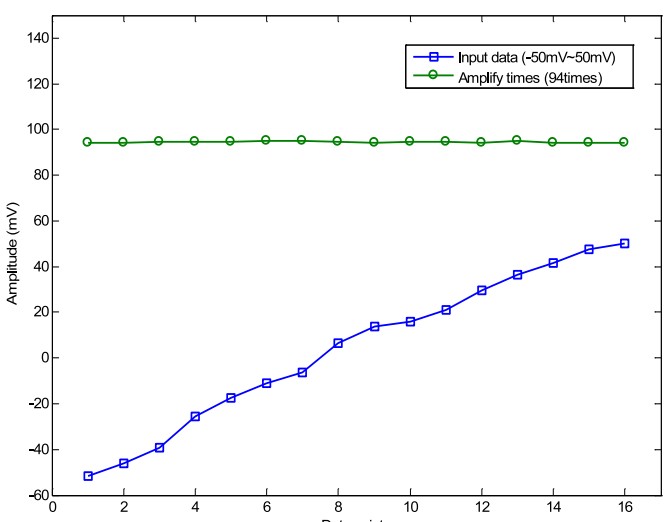

**Figure 11.** First Channel DC Test

5      Next is the conformance test between channels of the receiver. The test is to better maintain high consistency between channels. In the test, positive terminals of the six channels are connected together, while the negative terminals are connected with each other.The two connected terminals are input with standard sine wave.The amplitude of the input signal is 20mV peak-to-peak and the frequency is 150Hz. The extracted numerical values of each channel at the same time are compared to obtain the average error, the average absolute error and error percentage, as shown in Table 2.



**Table 2.** Error analysis table of inter-channel consistency

| Channel No. | Group1(mV) | Group2(mV) | Group3(mV) | Group4(mV) | Average(mV) | Average Error(mV) | Error Percentage(%) |
|---|---|---|---|---|---|---|---|
| Channel 1 | 20.7398 | 20.7372 | 20.7382 | 20.7359 | 20.7378 | 0.0020 | 0.1901 |
| Channel 2 | 21.0451 | 21.0465 | 21.0510 | 21.0465 | 21.0473 | 0.0022 | 0.3432 |
| Channel 3 | 20.7961 | 20.7927 | 20.7938 | 20.7964 | 20.7948 | 0.0013 | 0.1328 |
| Channel 4 | 21.0050 | 21.0044 | 21.0057 | 21.0015 | 21.0042 | 0.0008 | 0.2303 |
| Channel 5 | 21.0529 | 21.0504 | 21.0500 | 21.0530 | 21.0516 | 0.0013 | 0.0282 |
| Channel 6 | 20.8906 | 20.8911 | 20.8906 | 20.8914 | 20.8909 | 0.0004 | 0.2013 |

In the AC testing, the six channels of positive and negative terminals are connected in parallel, and the sine waves are input from the standard signal source. The output waveforms after the system acquisition are shown in Figure 12. The peak to peak value of input signal is 10mV, and the frequency is 20Hz. As can be seen from the figure, the consistency between each channel is quite high and no phase offset occurs. After the input signals are amplified by the system, peak to peak value displayed by waveforms of each channel reaches to 1000mV.

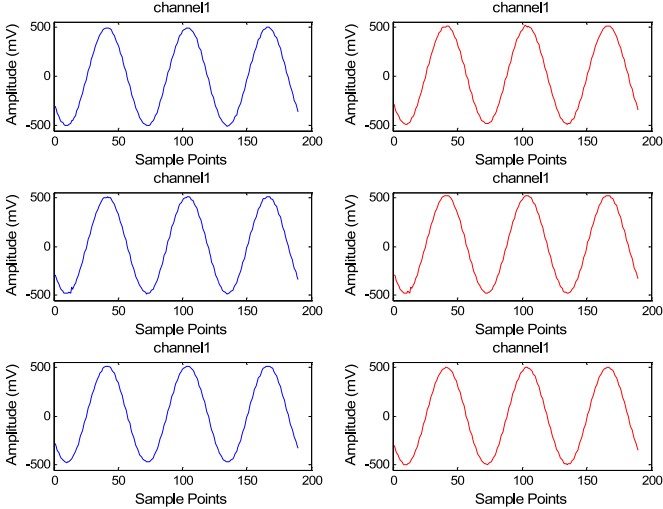

**Figure 12.** Standard sine wave of six channels in synchronous receiving

In Figure 13, the left figure is about the waveform of 3 cycles of data acquisition by the first channel shown in Figure 12. The right figure is about the frequency spectrum formed by waveforms in the left one, which underwent fast fourier transform. As is shown in the right one, the frequency of the input sines waves is 20Hz, with few harmonic components, which indicates the excellent performance of analog circuit board and high stability of power circuit.





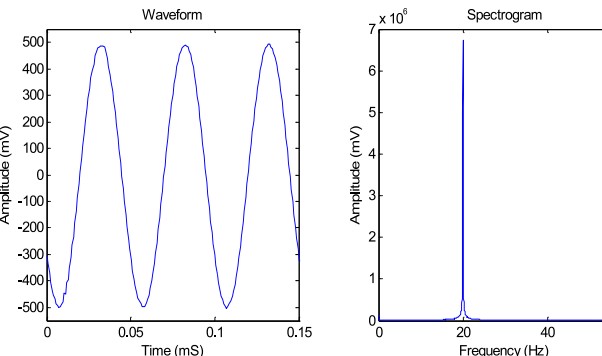

**Figure 13.** waveform and its spectrum diagram in AC test

## 5 Field testing

In the field testing, a hollow coil with 400turns and a diameter of 50cm is used as the test sensor, and its effective area is $40m^2$. The transmitter is Phoenix's T-4 transmitter, which transmits a bipolar pulsed waveform numbered TD50 with a 50% duty cycle(Wang et al.,2015).

5     Two sites were selected, one of which is a relatively open site while the other is the actual mining area. Near-field measurement method is used in the test. Grounded electrode source is adopted as the field source. Four aluminum plates as electrodes are respectively buried 40cm deep in the ground and covered with salt water and soil, so that the grounding resistance is less than 10 ohms. The distance between the transmitting electrodes A and B is 400 meters, and the four batteries are used as the power supply. The voltage ranges between 40-50V, while the emission current is 3A, and the emitting frequency is 25Hz.

10     During the test, the power supply electrodes are stationary, while the preset measurement line is parallel with the emission points A and B. The offset is 40 meters. The changing rate of the magnetic flux density (dB/dt) is received by the coil. Figure 14 below shows the one original one cycle.

    The open area where the test was carried out, is formed by the mixture of soil and slag. Each measurement line observed at 8 measuring points.After the data was stored, it was processed by computer program written in MATLAB and the secondary 15 field information of each measuring point was extracted and processed. The following profile curve was formed.

    Figure 15 shows the profile, which shows that the overall response tends to be flat. The signals received at the measurement points close to the transmitting nodes are relatively strong, while the central part of the profile becomes flat slowly.

    In order to test the performance of the receiver further, a mining area in Leshan, Sichuan Province was selected forthe field test, and a relatively gentle geodetic survey line was planned. The electrode source emission is adopted, and the distance 20 between the transmitting electrodes stance is 400 meters. The testing line offset is 80 meters, and the distance between the measuring points is 40 meters.



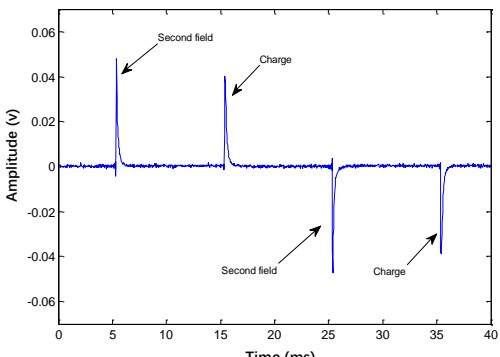

**Figure 14.** single cycle waveform

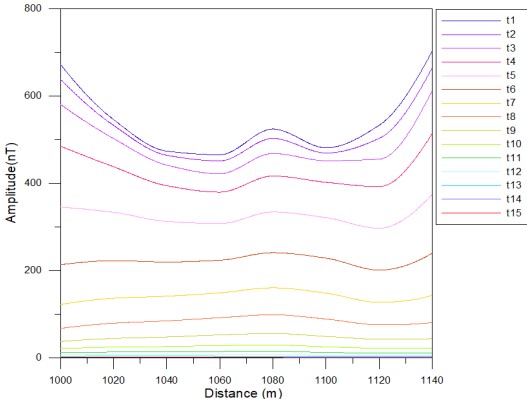

**Figure 15.** Close to the source L1 line profile

A continuous launch mode is used in mining area test, and the transmission signal frequency is 25Hz. Figure 16 is a waveform of two cycles of the raw data collected in the mining area.

After the collected data in the mining area are second-field extracted, filtered and interpolated, the waveform is formed. The waveform curve of the time domain is smoother after, 200 superimposed waveforms, which can well reflect the response of underground geological bodies to transient electromagnetic. The data of each measuring point are processed in the same way to obtain the pure secondary field curves, and the time domain order waveforms are extracted to form the profile of the measuring line. In Fig. 8, t1~t12 respectively represent the extraction time, and the extracted values of different collection points are collected together to form 12 curves at different times.

As can be seen by comparing figure18 and figure19 with figure17, the high anomaly point is near test point 1160, which happens to be a sloped metal vein. Thus it is verified that the receiver can acquire the signal of transient electromagnetic emission very well. By analyzing and comparing the profile with the actual geochemical distribution, it can be obviously seen





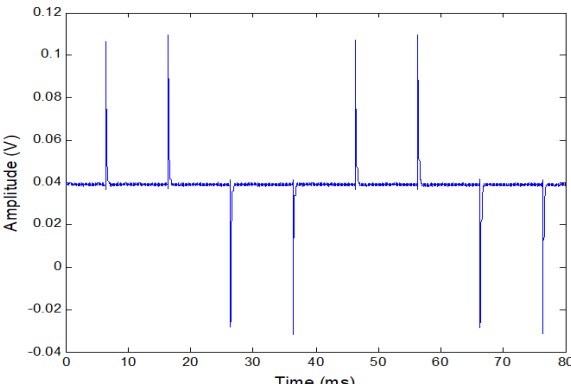

**Figure 16.** Waveform diagram acquired in mining area)

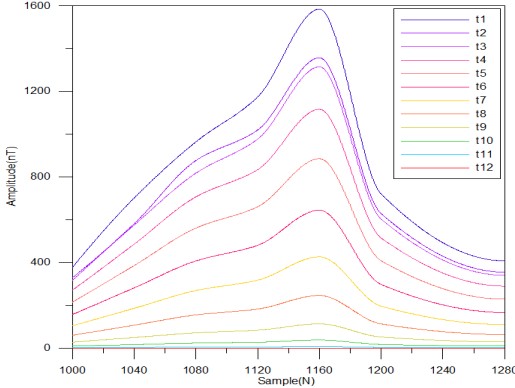

**Figure 17.** Survey line profile of copper mining area

that the abnormality of the curves is highly consistent near point 1160. In addition, the anomaly is consistent with the geological structure in this area.

## 6 Conclusions

The purpose of this paper is to develop a set of receiver device that can be applied in transient electromagnetic prospecting.

5 First, the hardware circuits and software programs are designed to realize all functions which are presented above. By means of dual controller, the instrument can receive signal synchronously through six channels. Then, the data stored in SD card are processed by computer programs to generate graphs. The overall performance of the receiver was tested and verified. All the collected data error of each channel is less than 0.35%, and, each channel can connect different sensors, such as coil, magnetic probe, electrode. This kind of receiver can be used to collect transient electromagnetic information. Due to its high

10 precision and high sampling rate, it can capture the fast falling edge of TEM, ultralow noise and so on. Hence, the multi-channel





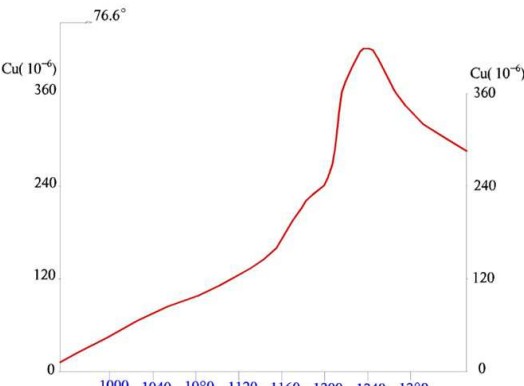

**Figure 18.** Anomaly of geochemical exploration of copper area along the survey line

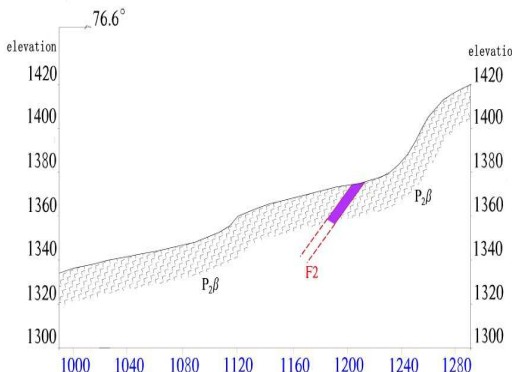

**Figure 19.** Geological profile of mining area

synchronous acquisition of magnetic field information in three directions and electric field information in two directions and the changing rate of magnetic induction intensity can be used for time domain reception. Meanwhile, the receiver can be used for pseudo random signal reception and distributed 3-dimension data reception, which can improve geophysical exploration efficiency.

5    *Code availability.*  The code of the receiver are available upon request(linfq@cdut.edu.cn)

*Data availability.*  The circuit schematics of the receiver are available upon request(linfq@cdut.edu.cn)



# Appendix A

## A1

*Author contributions.* FQ developed the main hardware and a part of software, XB instructed all the authors. And others participated in the experiments and software development.

5 *Competing interests.* The authors declare that they have no conflict of interest.

*Acknowledgements.* This work was supported by National key research and development project(no.2016YFC0600300),National natural science Foundation of China (NO.41674078),and Field observation and research base of ministry of land and resources of China for Geohazards in earthquake disturbed area of Longmen Mountain in Chengdu.





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
