# Peer review of "The development and test research of multi-channel Synchronous transient electromagnetic receiver"

_Geoscientific Instrumentation, Methods and Data Systems, 2018_

## Referee Comment (RC1) · T. Zhao (Referee) · 26 Apr 2018

GENERAL STATEMENT This paper designs a transient electromagnetic instrument with multi-channel synchronous receiving. Through data analysis, the measured data curves of the mining area is highly consistent with the existing geochemical exploration curves and geological profile. The research is innovative and reasonable. I suggest minor revisions of the manuscript to address the points raised in the specific. SPE-CIFIC COMMENTS 1. There are some grammatical errors in this article that need to be corrected. For example: In the abstract, "the measured data curves of the mining area is highly consistent" it should be" the measured data curves of the mining area

are highly consistent". 2ãĂĄApart from grammar, the error in the article content: (1)In the Multi-channels receiver hardware part, the article is more about the basic knowledge of hardware circuit design, not give the specific design steps of the experimental scheme. (2)In the Software design, such as STM32 software implementation part, A/D sampling, Flash storage, D/A output and Flash load, it is best to give the system flow chart, The design part of the hardware and software is not complete and clear. (3)In the figure4, not only do not detailed description of the waveform in the graph, but also do not explain the cause of the waveform, suggest elaborating on it in detail.

---

## Referee Comment (RC2) · Anonymous Referee #1 · 28 Apr 2018

The paper proposes a transient electromagnetic instrument with multi-channel synchronous and describes the experiment. They do a nice job of new TEM receiver. 1ïijL"PPS", "FPGA", "ARM" should be explain for first mention 2) There are minor typo errors: page1 line7 change "128 k" to "128 kHz" Page 5 line 11, change "second pulse signal" to"pulse per second", Page 8 change "40mS" to "40 ms". 3) Section 4 presents some results of the performance on the field. However, it would be interesting to see some field work photo and instrument picture. 4) the lab test is too simple, the main performance of the receiver contains band width, signal to noise ratio, self-noise level, input full scale, power consumption, linearity, et al. and the table 1 ,table 2, figure2, figure 3, figure 4 are not the key performance test. 5) Field test used Phoenix T-4 trans-

mitter, the authors should at least mention the performance of V8 receiver. And it may show the advantage of multi-channel TEM receiver. 6) In "introduction" and "Literature review" sections, there are too many words to describe the status in china, but it should be compare with international peer, and find the disadvantage of current instrument. The background and methodology and design have not been captured in detail. 7) The main innovation is multi-channel, add the result of filed test for supporting the advantage. 8) The description in the article is useful if the English is somewhat unclear. the English can be improved. 9) Could the authors discuss synchronous method in more detail?

Overall, the paper is well-written and presented. It proposes nice improvements to TEM receiver.

---

## Author Comment (AC1) · 10 May 2018

Dear Professor Zhao,

Thank you for your comments and precious suggestions on the manuscript! According to your advice, we revised this manuscript. All of the changes we have made are in the supplement file, which is marked-up manuscript version.

1. According to the comment, we have modified the paper thoroughly by correcting the grammatical errors, and also rephrased some paragraphs to present a clear expression. Such as: (1) In Page 1, Line 6: the sentence is changed as "...the

real-time synchronous transient electromagnetic acquisition system of six channels is achieved with 128k sampling rate". (2) In Page 2, Line 1: the sentence is change as" "...Conventional transient electromagnetic (TEM) uses mostly magnetic source or electrode source mode. In the magnetic source mode TEM volume effect is relatively small, and it has high detection accuracy." More modifications regarding grammatical errors are enclosed in the supplement file.

2. We devised a detailed design circuit diagram of the receiver and added specific design steps which are presented by eight figures (Figure2-9) in the manuscript, and gave detailed descriptions for these figures, which can be seen in the supplement file.

3. We have added the flowchart (Figure 10) in the manuscript for the system program, which makes the software design complete and clear.

4. We have described the characteristics of the waveform in Figure 4, and explained the cause of the waveform in the discussion version of the manuscript. Now, Figure 4 in the original version of the manuscript has been changed as Figure 13 in the discussion version. We elaborated Figure 13, and added "the left figure is about the waveform of 3 cycles of data acquisition by the first channel shown in Figure 12. The right figure is about the frequency spectrum formed by waveforms in the left one, which underwent fast fourier transform. As is shown in the right one, the frequency of the input sine waves is 20Hz, with few harmonic components, which indicates the excellent performance of analog circuit board and high stability of power circuit".

Once again, thank you very much for your comments and advice. We have tried our best to modify and make all necessary changes in the manuscript. Meanwhile, we are deeply grateful for the time and efforts of the editors and reviewers, and we sincerely hope that these amendments will meet your expectation.

Please also note the supplement to this comment:
https://www.geosci-instrum-method-data-syst-discuss.net/gi-2018-6/gi-2018-6-AC1-

supplement.zip

---

## Author Comment (AC2) · 15 May 2018

Dear Anonymous Referee: Thank you for your time and very useful comments! According to your advice, we revised this manuscript. All of the changes are made in the supplement files, which are a marked-up manuscript version and a clear revised version.

1. Comment from Reviewer: "PPS", "FPGA", "ARM" should be explain for first mention. Author's response in manuscript: In Page 8 Line 7, the abbreviation of PPS means pulse per second, and in Page 2, Line 15, FPGA means Field Programmable Gate Array, and ARM means Advanced RISC (reduced instruction set computer) Machine.

[Figure]

These explanations were provided in the marked-up manuscript version.

2. Comment from Reviewer: There are minor type errors: Page1 Line7 change "128 k" to "128 kHz" Page 5 Line 11, change "second pulse signal" to" pulse per second", Page 8 change "40mS" to "40 ms". Author's response in manuscript: We have modified the typing errors in the manuscript. In Page 1 Line 7: 128k refers to the sample rate, and it should be changed to 128k SPS (Sample Per Second). In Page5 Line 11: the PPS means Pulse Per Second. In Page 13, Figure 15: the period of the emitting signals is 40mS, and the "S" in "40mS" should be lowercase. 40mS was changed to 40ms.

3. Comment from Reviewer: Section 4 presents some results of the performance on the field. However, it would be interesting to see some field work photo and instrument picture. Author's response in manuscript: we added photos of the instrument and the field work, which are shown in Figure 21, the photo of the receiver, Figure 22 T-4, photo of the transmitter in operation and Figure 23, the field working photo of exploration area. Those pictures were inserted after the conclusion.

4. Comment from Reviewer: the lab test is too simple, the main performance of the receiver contains band width, signal to noise ratio, self-noise level, input full scale, power consumption, linearity, et al. and the table 1 ,table 2, figure 2, figure 3, figure 4 are not the key performance test. Author's response in manuscript: we have given the following statements in response to the reviewer's comment. (1) In Page 5, Line 4, the band width of the receiver is DC to 12.8k SPS, (2)In Page 11, Line 1: SNR=20*log (Full Scale Value/Root-Mean-Square Noise Voltage, and the signal-to-noise ratio of 6 channels is approximately 100dB by calculating. (3)In Page 4, Line 3: the multi-channel receiver has a wide voltage input range, which varies from -5V to +5V, and the amplitude of the signal in actual test is at the millivolt level. In Page 4, Line 9, the dynamic range of analog-digital converter is up to 144dB, because the resolution of the analog to digital convertor is a 24-bit resolution. (4) In Page 7 Line 6, since all chips are low-powered, and the overall power consumption of the receiver is lower than 10 watts. (5) In the manuscript, Table 1 and Table 2 show the amplification

ratios of each channel under the same input condition. From table 2 we can see the average error is less than 2.2$\mu$V. All of these changes were made in supplemented file. (6) In Page 12 Line 3: the noise of the circuit itself is collected when each channel of the receiver is connected to zero. And Figure 14 was added to present it. The figure can be seen in supplement file.

5. Comment from Reviewer: field test used Phoenix T-4 transmitter, the authors should at least mention the performance of V8 receiver. And it may show the advantage of multi-channel TEM receiver. Author's response in manuscript: In Page 12 Line 5: we added the following description. Phoenix T-4 transmitter is a small power transmitter. This transmitter powered by battery group can launch up to 40A current, and it can launch many kinds of waveforms, such as TD50 (named by Phoenix), which has a duty ratio of 50%. V8 receiver is the eighth generation of receiver technology developed by Phoenix since 1975. The sample rate is 96k, while the memory card used in V8 receiving system is only 512M bytes, which can't store all received data. V8 system is synchronized by GPS time. But the sample rate of the multi-channel TEM receiver proposed in the manuscript is 128k SPS, and synchronization mechanism is realized by GPS module and 28-bits counter. The synchronization accuracy can be improved greatly, and all the time information and data information are stored in SD card simultaneously.

6. Comment from Reviewer: in "introduction" and "Literature review" sections, there are too many words to describe the status in china, but it should be compare with international peer, and find the disadvantage of current instrument. The background and methodology and design have not been captured in detail. Author's response in manuscript: we have modified the part of "introduction" and "literature review", and the two parts were merged into one part named "Introduction". "Transient electromagnetic (TEM) method belongs to the active field source method of time domain electromagnetic. J.R.Wait firstly proposed the use of transient electromagnetic method to search for conducting ore bodies in 1951. In recent decades, the TEM receiver has been widely used in metal mineral, petroleum, natural gas exploration and other fields (Danielsen et al., 2003; Haroon et al., 2015), so many kinds of TEM receiver have been developed and manufactured, such as V8 receiver from Phoenix Geophysics(Phoenix Geophysics, 2017), ADU-07e from Metronix (Metronix, 2017), KMS-820 from KMS(KMS, 2017), which are easy to use and have good performance. Meanwhile, the TEM detection theories have also been developed. In 2010, internationally renowned geophysicist Zhdanov proposed the following directions for future electromagnetic exploration instruments and methods in the 75th anniversary of "Geophysics": Multi-component emission, Multi-channel reception and Pseudo-seismic data collection (Qi et al., 2015, Ayuso et al., 2016). Xue Guoqiang proposed short offset transient electromagnetic (SOTEM) method (Xue et al., 2013, Chen et al.,2016, Chen et al.,2017), and the reduction of transceiver distance can both greatly enhance the exploration depth and improve the signal to noise ratio (SNR). Because of the need for deeper exploration depth, and the need to obtain more diverse subsurface geological information, multi-channel number, low SNR and high synchronization are the key elements for TEM receiver development, which are useful to obtain more abundant underground geological information and improve the accuracy of detection. Therefore, we propose a multi-channel synchronous transient electromagnetic receiver, which has the characteristics of multi-channel synchronous parallel acquisition, large capacity (64G) storage function with full time synchronization and the sampling rate up to 128k SPS, while the sampling rate of V8 receiver is 96k only. The main purpose of this paper is to introduce a receiver system for synchronous acquisition of multiple electromagnetic signals in transient electromagnetic prospecting to achieve multi-parameter and multi-channel synchronous reception. High-speed programmable logic devices are used to achieve high-level synchronization between channels. Transmitting current waveform acquisition and multi-channel reception can be synchronized by using high-precision Global Position System(GPS) timing unit which is controlled by serial port of micro controller, while a programmable high-precision counter is used to store another data information synchronously, which is adopted to further improve the synchronization

accuracy when the receivers work in distributed mode."

7. Comment from Reviewer: the main innovation is multi-channel, add the result of field test for supporting the advantage. Author's response in manuscript: The receiver has the following innovations: firstly, the channel number of the receiver can be flexibly configured as even number. In laboratory testing, the receiver is configured as six-channel receiving system, and Figure 12 and 14 display signal data that is collected synchronously. Secondly, the high precision synchronization method is realized by GPS module and high resolution counter together. Thirdly, the receiver has high ripple rejection ratio, which is up to 82dB, and low power consumption by the application of high performance LDO devices. Fourthly, the receiver implements real time information storage, and the synchronization sampling rate is as high as 128k SPS. But in mine testing, due to the inconvenience of the layout of multi electrode pairs and coils, the data receiver and current waveform receiver are configured as two channels, while one channel of current waveform receiver is used to receive currents. Three waveform figures are shown in Figure 17. The first sub-figure is about the change rate of magnetic field(dB/dt), and the second one is about the electric field intensity Ex, and the third one is about the transmitting current waveform. Synchronous receiver with 32 channels can be realized, according to the bandwidth of the storage speed of SD card.

8. Comment from Reviewer: the description in the article is useful if the English is somewhat unclear. The English can be improved. Author's response in manuscript: We have tried our best to improve the English expression and corrected the grammatical errors in the manuscript. And the changes will be seen in the supplement file.

9. Comment from Reviewer: could the authors discuss synchronous method in more detail? Author's response in manuscript: In Page 13 Line 10, the high-precision synchronization mechanism is implemented with a high reliability GPS module and a 28-bit counter in FPGA. The receiver's time information is refreshed every two seconds by PPS of GPS module. The 28-bit counter is used to record the pulse, which come from 25MHz clock in FPGA board. The 28-bit counter value and collecting data and

time information are packaged and stored in SD card. The error for each count value is 40ns. Each sample data point can be tracked by the stored time information and data information in SD card. After data collection, no matter how many receivers work in mine area at the same time. The full-time range store technology supplied a valuable method to enhance the accuracy of the synchronization.

We look forward to hearing from you regarding our latest submission. We would be glad to respond to any further questions and comments that you might have. Thanks again for your valuable suggestions!

Please also note the supplement to this comment:
https://www.geosci-instrum-method-data-syst-discuss.net/gi-2018-6/gi-2018-6-AC2-supplement.pdf

———————————————————

[Figure]

**Fig. 1.** Figure 21 Receiver

[Figure]

**Fig. 2.** Figure 22 T-4 transmitter

[Figure]

**Fig. 3.** Figure 23 Mine Picture

**Supplement:**

**The development and test research of multi-channel **Synchronous** synchronous transient electromagnetic receiver**

Fanqiang Lin1,2, Xuben Wang2,3, Kecheng Chen1, Depan Hu4, Song Gao1, Xue Zou1, and Cai Zeng1 1College of Information Science and Technology, Chengdu University of Technology, Chengdu 610059, China 2College of Geophysics, Chengdu University of Technology, Chengdu 610059, China 3Key Lab of Geo-Detection and Information Techniques of Ministry of Education, Chengdu 610059, China 4Geoenvironment Monitoring Institute in Chengdu, Chengdu 610071, China **Correspondence:** Fangiang Lin(linfq@cdut.edu.cn)

**Abstract.** As a result of the drastic reduction of shallow mineral resources, the exploitable potential and reserve of proven mines have been provenare insufficient, and the mineral resources in deep ground need to be more refinedly explored. Because There are some disadvantages of the existing instrumentshave less channels and insufficient, such as few channels and slow sampling rate, the development and experiment of the etc. Therefore, multi-parameter transient electromagnetic instrument

- 5 with multi-channel synchronous receiving are therefore proposed synchronous receiving has been developed and tested. The instrument is composed of two controllers, embedded controller and programmable logic controller, which can provide diversified information combination for the following information processing. Under the grounding electrode source emission mode, the real-time synchronous transient electromagnetic acquisition system in six channels with 128k sampling rate is achieved of six channels is achieved with sampling rate of 128k SPS(Sample Per Second). The data of the six channels are recorded in the
- 10 full time range in the time domain. At the same time, experiments were carried out in the laboratory, open areas and actual mine. Through data analysis, the measured data curves of the mining area is are highly consistent with the existing geochemical exploration curves and geological profile.

**1 Introduction**

Transient electromagnetic (TEM) method belongs to the active field source method of time domain electromagneticmethod.

- 15 The principle is that the conductive geological body existing underground produces induced eddy currents under the action of alternating electromagnetic fields, and the eddy current further generates secondary magnetic fields. In transient electromagnetic method, it often using periodic bipolar waves as the launch signal, data acquisition is completed during the bipolar wave intermittent, thus effectively avoiding the coupling problem in the frequency domain electromagnetic method. The instrument collects and studies characteristics of the intensity, distribution, space and time decay of the second field, and then deduces the
- 20 underground anomalies. The transient electromagnetic signals contain very rich frequency components, which are very useful for low-resistivity geologic bodies with high penetration, which can effectively improve the actual exploration depth. Conventional transient electromagnetic (TEM) use mostly magnetic source or electrode source mode, the magnetic source mode TEM volume effect is relatively small, high detection accuracy, but the effective detection depth is usually less than 500 meters, and

the high Geological response is not sensitive. However, LOTEM can increase the exploration depth up to 10 km. However, due to the large distance between the LOTEM transmission and reception, the accuracy of the collected data is relatively low, as well as a poor signal-to-noise ratio and increased possibility of interference on the way of long-distance transmission(Brunke et al., J.R. Wait firstly proposed the use of transient electromagnetic method to search for conducting ore bodies in 1951. In

- 5 recent decades, the TEM receiver has been widely used in metal mineral, petroleum, natural gas exploration and other fields (Danielsen et al., 2003; Haroon et al., 2015), so many kinds of TEM receiver have been developed and manufactured, such as V8 receiver from Phoenix Geophysics(Phoenix Geophysics, 2017). As a result, domestic scholars such as Xue Guoqiang proposed short offset transient electromagnetic (SOTEM)device method(Xue et al., 2013,Chen et al., 2016,Chen et al., ADU-07e from Metronix (Metronix, 2017), KMS-820 from KMS(KMS, 2017), the reduce of transceiver distance can both greatly enhance the
- 10 exploration depth, but also effectively improve the signal to noise ratio. In recent years, it has a very broad application prospect in such fields as engineering environment, geological disaster investigation, underground gob area, groundwater, petroleum, coal and other non-metallic mineral exploration and advanced geological prediction.

**2 Literature review**

which are easy to use and have good performance. Meanwhile, the TEM detection theories have also been developed. In

- 15 2010, internationally renowned geophysicist Zhdanov proposed the following directions for future electromagnetic exploration instruments and methods in the 75th anniversary of "Geophysics": Multi-component emission, Multi-channel reception and Pseudo-seismic data collection (Qi et al., 2015, Ayuso et al., 2016). As the demand for resources and energy increases due to economic development, the depth is increased to the second depth space (500-2000m) on the basis of the existing exploration depth (500m depth range). The fine exploration within this range will be a long-term and heavy task to study
- 20 new theories and methods and to develop equipment with higher adaptability to make up the lack of traditional exploration methods and instruments. China's overall theoretical level and engineering practice in the field of transient electromagnetic remain at the same level with the international, but the domestic instruments in the technical indicators comparing with foreign countries there is still a certain gap, which will be constraining China's original innovation and quantitative calculation of the bottleneck(Zhong Xue Guogiang proposed short offset transient electromagnetic (SOTEM) method (Xue et al., 2013, Chen et
- 25 al.,2016, Zhang-Chen et al.,2017). With the development of electrical prospecting theory, instrument's design and development, many kinds of instrument products have been made. The main development directions are automation, intelligence, refinement and lightness. As a result, various institutions of higher learning and research institutes in China have introduced various forms of transient electromagnetic prospecting instruments in their decades of development. By comparing most of the transient electromagnetic surveys at home and abroad since recent 20 years, the main problems and gaps in the domestic transient
- 30 electromagnetic instrument lie in that: some technical indicators are lagging behind; the function is insufficient; the consistency is low, and so on. It can be seen that the development of new acquisition devices, the reduction of transceiver distance can both greatly enhance the exploration depth and improve the signal to noise ratio (SNR). Because of the transformation of new launch modes, acquisition methods, need for deeper exploration depth, and the development of new instruments and

evaluation methods of electromagnetic methods, etc. are particularly important(Li et al.,2012). The core need to obtain more diverse subsurface geological information, multi-channel number, low SNR and high synchronization are the key elements for TEM receiver development, which are useful to obtain more abundant underground geological information and improve the accuracy of detection. Therefore, we propose a multi-channel synchronous transient electromagnetic receiver, which has

5 the characteristics of multi-channel synchronous parallel acquisition, large capacity (64G) storage function with full time synchronization and the sampling rate up to 128k SPS, while the sampling rate of V8 receiver is 96k only.

The main purpose of this paper is to develop introduce a receiver system for synchronous acquisition of multiple electromagnetic signals in transient electromagnetic prospecting to achieve multi-parameter and multi-channel synchronous reception. High-speed programmable logic devices are used to achieve high-level synchronization between channels. Waveform

10 Transmitting current waveform acquisition and multi-channel receive reception can be synchronized by using high-precision GPS timing unit to synchronize, combined with Global Position System(GPS) timing unit which is controlled by serial port of micro controller, while a programmable high-precision counter to is used to store another data information synchronously, which is adopted to further improve the synchronization accuracy - when the receivers work in distributed mode.

**2 Multi-channels Multi-channel receiver hardware and software design**

**15 2.1 Receiver framework**

Receiving system consists of four parts, namely: FPGA(Field Programmable Gate Array) unit, ARM(Advanced RISC Machine) unit, analog acquisition unit and power supply circuit. System block diagram shown in Figure 1. Analog board channel numbers are The numbers of analog board are marked as channel 1/2, 3/4, 5/6 channels respectively, respectively, and each analog board have two channels. The board contain contains the signal conditioning circuit, amplification and acquisition of all channels

20 which are completely independentand, and multi-channel real-time synchronized acquisition is achieved in multi-channel.

Figure 1. System block diagram of hardware

**2.2 Design of analog circuit board**

Analog board mainly includes signal conditioning (including: the first stage instrumentation amplifier, the second stage programmable amplifier, low-pass filter), protection circuit (positive and negative polarity of each input stage clamp diode), the natural potential compensation circuit(Liu et al.,2016), single-end to differential circuit and analog to digital converter.

5

The first stage amplifier uses a dedicated high-precision, low-noise INA114, and using the AD5272 high-resolution digital potentiometer is used to design the precision gain op-amp, to achieve high-precision amplification and range adjustment. The two circuits are shown in Figure 2 and Figure 3.

Figure 2. INA114 Instrument Amplifier and Programmable Gain Amplifier Circuit

It utilizes the PGA103 program-controlled amplifier, making use of Programmable Gain Amplifier, and a general IO port to 10 complete the three level amplification ratio adjustment.

Figure 3. AD5272 Digital potentiometer

---

## Referee Comment (RC3) · Anonymous Referee #1 · 1 Jun 2018

Authors have made a serious answer to my questions. So I recommend that this paper can be accepted for publication.
* * *

---

## Referee Comment (RC4) · T. Zhao (Referee) · 4 Jun 2018

Dear Authors:After I checking the answers of my 4 questions, so I recommend that this paper can be accepted for publication. Good luck!